# Insights into the Gut Microbiome of the South American Leaf-Toed Gecko (*Phylodactylus gerropygus*) Inhabiting the Core of the Atacama Desert

**DOI:** 10.3390/microorganisms12061194

**Published:** 2024-06-13

**Authors:** Daniela S. Rivera, Valentina Beltrán, Ignacio Gutiérrez-Cortés, Constanza Vargas, Fernando D. Alfaro

**Affiliations:** 1GEMA Center for Genomics, Ecology & Environment, Universidad Mayor, Camino La Piramide, 5750, Santiago 8580745, Chile; valentina.beltranz@mayor.cl; 2Extreme Ecosystem Microbiomics & Ecogenomics Lab., Facultad de Ciencias Biológicas, Pontificia Universidad Católica de Chile, Santiago 8320165, Chile; igutierrezc@estudiante.uc.cl; 3Centro UC Desierto de Atacama, Instituto de Geografía, Pontificia Universidad Católica de Chile, Vicuña Mackenna 4860, Santiago 7820436, Chile; cgvarga1@uc.cl

**Keywords:** gut microbiome, *Phyllodactylus gerropygus*, Atacama Desert

## Abstract

Living in arid environments presents unique challenges to organisms, including limited food and water, extreme temperatures, and UV exposure. Reptiles, such as the South American leaf-toed gecko (*Phyllodactylus gerrhopygus*), have evolved remarkable adaptations to thrive in such harsh conditions. The gut microbiome plays a critical role in host adaptation and health, yet its composition remains poorly characterized in desert reptiles. This study aimed to characterize the composition and abundance of the gut microbiome in *P. gerrhopygus* inhabiting the hyperarid Atacama Desert, taking into account potential sex differences. Fecal samples from adult female and male geckos were analyzed by 16S rRNA gene amplicon sequencing. No significant differences in bacterial alpha diversity were observed between the sexes. However, the phylum Bacteroidota was more abundant in females, while males had a higher Firmicutes/Bacteroidota ratio. The core microbiome was dominated by the phyla Bacteroidota, Firmicutes, and Proteobacteria in both sexes. Analysis of bacterial composition revealed 481 amplicon sequence variants (ASVs) shared by female and male geckos. In addition, 108 unique ASVs were exclusive to females, while 244 ASVs were unique to males. Although the overall bacterial composition did not differ significantly between the sexes, certain taxa exhibited higher relative abundances in each sex group. This study provides insight into the taxonomic structure of the gut microbiome in a desert-adapted reptile and highlights potential sex-specific differences. Understanding these microbial communities is critical for elucidating the mechanisms underlying host resilience in Earth’s most arid environments, and for informing conservation efforts in the face of ongoing climate change.

## 1. Introduction

Organisms have developed remarkable adaptations to survive in extreme environments such as deserts, polar regions, deep oceans, and high altitudes [1,2]. These environments present unique challenges, including extreme temperatures, limited food and water resources, high or low pressure, high salinity, low oxygen levels, and acidic or basic pH levels [1,3]. Some vertebrate groups have successfully inhabited arid regions, which harbor high levels of reptile diversity. These reptiles have evolved unique adaptations, including changes in reproduction, metabolism, immunity, and behavior [4,5,6]. The gut microbiome of desert reptiles also plays a crucial role in their adaptation, health, and, potentially, their response to environmental changes [7,8,9,10]. The gut microbiome consists of a dynamic and balanced microbial community in the host organism and plays an essential role in the host’s nutrition, metabolism, and immunity against potential pathogens [10,11]. Previous studies have shown that desert environments, including temperature and drought, can alter the composition of the gut microbiome of reptiles, potentially impacting host health and fitness [5,12]. However, vertebrate gut microbial communities can also be influenced by several other factors, such as host features (e.g., age and sex) [13,14,15,16,17]. For instance, research on striped plateau lizards (*Sceloporus virgatus*) revealed that female lizards tend to harbor more diverse gut microbiomes than males [13]. This suggests potential disparities in energy allocation, activity levels, and feeding behaviors between the sexes [13,18].

Geckos stand out among reptiles for their remarkable diversity in ecological features, showcasing a wide array of morphological and behavioral adaptations [19,20]. They constitute a hyper-diverse, ancient, and globally distributed group [21]. Many gecko species are nocturnal, foraging at night to conserve energy and avoid the intense desert sun [21,22], a behavior that also aids in predator avoidance. Compared to other lizards with similar activity times, geckos exhibit lower body temperatures and prefer lower temperatures [21]. Like other reptiles, geckos have adapted to survive on very little water supply, obtaining most of the water they need from their prey [23,24,25]. Some species, like fat-tailed geckos, can store up to nine months’ worth of food in their tails and subsist on very little water [26,27]. Thick skin protects geckos from the harsh elements of their desert environments [27,28]. While most geckos are predominantly carnivorous, many are also omnivorous and, to a lesser extent, herbivorous [21]. Geckos have recently been classified as sit-and-wait predators [22,29]. However, there is an emerging view of geckos as active, olfactory predators capable of sustained locomotion over long distances [30,31].

The South American leaf-toed gecko (*Phyllodactylus gerrhopygus*) is a typical oviparous animal with crepuscular and nocturnal habits, distributed from Peru to Chile [32,33,34,35]. It is the only *Phyllodactylus* species inhabiting Chile [32,36,37]. It is found in the extreme north of Chile, from sea level to 3500 masl [37,38], although there are few sectors where it exceeds 1000 masl [32,37]. It inhabits diverse environments, from sandy beaches to rocky ravines in mountainous sectors and extremely arid desert areas. It can be observed under marine vegetation deposited on beaches, under stones in areas of extreme aridity, and in ravines with shrub and/or tree vegetation. This species is the top predator in the *Tillandsia* fields, a unique fog-dependent ecosystem across the Atacama Desert [32,39]. The South American leaf-toed gecko is mainly insectivorous, with its diet characterized by the consumption of arthropods, mainly Coleoptera and insect larvae [39], which are essential energy sources providing nutrients such as carbohydrates, lipids, and protein [40]. It presents a foraging strategy intermediate between active and ambush foraging, involving the active search for some of its prey. Occasional consumption of plant material has also been reported [39]. In Chile, the South American leaf-toed gecko has been categorized as vulnerable according to the Hunting Law (DS 5/1998 MINAGRI, Reglamento de Caza 2011 SAG), with its area of application being the North (from Arica and Parinacota to Atacama). Ecologically significant, in terms of trophic position, *P. gerrhopygus* consumes some resources that allow it to occupy the top of the food web, even surpassing some species of lizards such as *Liolaemus reichei* [41].

The environmental stressors prevalent in arid habitats, such as limited food and water availability, extreme temperatures, and direct exposure to UV radiation, could profoundly influence microbial communities [42]. This underscores the critical role these microorganisms play in the development and health of their host organisms, particularly in adapting to arid conditions. As climate change threatens arid ecosystems and their inhabitants worldwide, it is increasingly important to comprehensively characterize the functional and taxonomic composition of microbial communities thriving in these environments. Such knowledge promises to elucidate the mechanisms underlying organisms’ resilience in the planet’s most arid regions. In this context, desert reptiles provide invaluable model organisms for studying gut microbiomes, offering a unique opportunity to explore the potential mechanisms driving survival and persistence under hyperarid conditions. Therefore, this study aimed to characterize the composition and abundance of the gut microbiome of wild-caught South American leaf-toed geckos in a sex-dependent manner that inhabit the hyperarid Atacama Desert.

## 2. Methodology

### 2.1. Site Information

The study site is situated in northern coastal Chile at the Estación Atacama UC Oasis de Niebla Alto Patache (20°49′ S–70°09′ W), located on the summit of the Coastal Cordillera (850 m above sea level) about 3.5 km linear distance from the coastline on a peninsula oriented opposite the coastal southwest wind direction. The regional climate is hyperarid, with a mean annual precipitation of approximately 0.96 mm and a mean annual temperature of 18.5 °C (for the last 43 years, Dirección Meteorológica de Chile, 2024). Fog cover is frequent year-round but is most intense during the winter and spring months from June to October [43,44]. Moisture from marine fog is essential to maintaining an isolated and diverse ecosystem.

### 2.2. Animal Trapping

Initially, we captured geckos over 3–4 consecutive days in September 2020. Adult (snout–vent length = 49 ± 5 mm) female (n = 8) and male (n = 8) geckos were trapped, and field researchers were trained in proper handling techniques to safely capture geckos by hand without causing harm. Geckos with a snout-vent length of less than 45 mm were classified as juveniles [45] and excluded in this study.

Our hand-capture surveys were conducted with a strong focus on vegetation and rocky surfaces, both during the day and at night. To capture ground-dwelling geckos, we installed pitfall traps, which were regularly checked to minimize stress and mortality. We recorded the sex of all captured animals, noting that male geckos may have a larger bulge just caudal to their cloaca, while females do not [46]. Each animal was captured only once, and once the necessary sample was obtained, they were released back to the exact location of capture. All procedures for handling live animals were approved by the Servicio Agrícola y Ganadero (SAG), Chile (5497/2019 and 342/2020), and every effort was made to minimize animal suffering.

### 2.3. Microbiome Analysis

#### 2.3.1. Stool Sample Collection and DNA Isolation

Fecal matter was collected directly from the anus using sterile FLOQSwabs™ (CE 0123, COPAN, ITALIA, spa, Brescia, Italy), immediately placed in Eppendorf tubes containing 500 μL of RNAlater, and stored at 4 °C until DNA extraction. For the DNA extraction, we used the PowerFecal Pro kit (QIAGEN GmbH, Hilden, Germany) according to the manufacturers’ instructions. The isolated DNA samples were sent to the Zymo Research Central Laboratory (Irvine, CA, USA) for further analysis.

#### 2.3.2. Targeted Library Preparation

The DNA samples were prepared for targeted sequencing with the Quick-16S™ Plus NGS Library Prep Kit (Zymo Research, Irvine, CA, USA). These primers were custom-designed by Zymo Research to provide the best coverage of the 16S gene while maintaining high sensitivity. The primer sets used in this project were Quick-16S™ Primer Set V3-V4 (Zymo Research, Irvine, CA, USA).

The sequencing library was prepared using an innovative library preparation process in which PCR reactions were performed in real-time PCR machines to control cycles and limit PCR chimera formation. The final PCR products were quantified with qPCR fluorescence readings and pooled together based on equal molarity. The final pooled library was cleaned up with the Select-a-Size DNA Clean & Concentrator™ (Zymo Research, Irvine, CA, USA), then quantified with TapeStation^®^ (Agilent Technologies, Santa Clara, CA, USA) and Qubit^®^ (Thermo Fisher Scientific, Waltham, MA, USA).

#### 2.3.3. Control Samples

The ZymoBIOMICS^®^ Microbial Community Standard (Zymo Research, Irvine, CA, USA) was used as a positive control for each DNA extraction, if performed. The ZymoBIOMICS^®^ Microbial Community DNA Standard (Zymo Research, Irvine, CA, USA) was used as a positive control for each targeted library preparation. Negative controls (i.e., blank extraction control, blank library preparation control) were included to assess the level of bioburden carried by the wet-lab process.

#### 2.3.4. Sequencing and Bioinformatics Analysis

The final library was sequenced on Illumina^®^ NextSeq 2000™ (Zymo Research, Irvine, CA, USA) with a p1 (cat 20075294) reagent kit (600 cycles). The sequencing was performed with a 30% PhiX spike-in. Unique amplicon sequences were inferred from raw reads using the Dada2 pipeline [47]. Chimeric sequences were also removed with the Dada2 pipeline. Taxonomy assignment was performed using Uclust from Qiime v.1.9.1. Taxonomy was assigned using the Zymo Research Database, a 16S database that is internally designed and curated, as a reference. Composition visualization, alpha-diversity, and beta-diversity analyses were performed with Qiime v.1.9.1 [48].

### 2.4. Statistical Analysis

All data are presented as the mean ± standard error (SEM). The species richness of bacterial communities was estimated using ACE and Chao1 indices, while bacterial diversity was assessed using Shannon diversity (H′) and Simpson diversity indices. The differences in bacterial composition between female and male geckos were evaluated by a one-way PERMANOVA test using Bray–Curtis distance matrix. We used both *t*-tests and non-parametric analysis (i.e., Mann–Whitney) to analyze the data, depending on whether the data met the assumptions of normality. Normality assumptions were confirmed using the Shapiro–Wilk test. All statistical procedures were performed using R software version 4.0.0 (R Development Core Team, 2020, Vienna, Austria). Each sample was rarefied to the same sequencing depth so that the number of artifacts that appear in each sample is controlled. We rarefied the samples using the R package Microeco version 1.5.0 [49]. Differences were considered statistically significant at *p* < 0.05.

## 3. Results

### 3.1. Diversity of the Gut Bacterial Communities Was Not Significantly Different between Female and Male Geckos

To characterize bacterial richness, rarefaction analysis was performed by randomly sampling 1000 times with replacement and estimating the total number of Amplicon Sequence Variants (ASVs) present in female and male gecko samples. The curve in each group reached saturation in most of the samples, indicating that the sequencing data was great enough that very few new ASVs remained undetected (Appendix A). Analysis using a *t*-test indicated no significant differences in alpha diversity between sexes (H′ index: *p* = 0.08 and Simpson indices: *p* = 0.52; Figure 1A,B). Similarly, ACE and Chao1 richness indices showed no significant differences between female and male geckos (both *p* = 0.32; Figure 1C,D), although we observed a higher number of ASVs in males.

### 3.2. Taxonomic Characterization of the Gut Bacterial Communities in Female and Male Geckos

A total of 833 ASVs were recorded across female and male geckos (of which 589 ASVs corresponded to samples of female animals and 725 ASVs corresponded to male animals). Sequence analysis showed all the ASVs from female and male geckos were classified into thirteen phyla, of which the most dominant phylum in fecal samples was Bacteroidota (41.52%), Firmicutes (41.45%), and Proteobacteria (13.60%) (Figure 2A). Actinobacteria, Cyanobacteria, Tenericutes, Verrucomicrobia, Deferribacteres, Elusimicrobia, Fusobacteria, Gemmatimonadetes, Planctomycetes, and Saccharibacteria were minor components and were not present in all samples.

Comparison between female and male samples revealed a similar bacterial phyla abundance, except for a higher abundance of Bacteroidota in females (Student’s *t*-test = 2.21; *p* = 0.04; Figure 2B), with the class Bacteroidia (family Bacteroidaceae) being the significantly most predominant (Figure 2C,E). At the gender level, *Bacteroides* was the most abundant (Figure 2F). Another family that showed abundance in females, although not significantly different from males, was Porphyromonadaceae, represented by the genus *Parabacteroides* (Figure 2E,F). In male geckos, we observed a relative increase in the ratio of Firmicutes to Bacteroidota, which was approximately 2.0 times higher than in females (Mann–Whitney U = 8.00; z = −2.47; *p* = 0.01; Figure 2 and Figure 3). In both female and male geckos, Firmicutes were mostly represented by the class Clostridia, order Clostridiales, and at the family level by Lachnospiraceae and Ruminococcaceae. At the genus level, *Lachnoclostridium*, *Anaerotruncus*, and *Oscillibacter* (Figure 2C–F). In Proteobacteria, the class Deltaproteobacteria, order Desulfovibrionales, family Desulfovibrionaceae, and genus *Desulfovibrio* were the most representative in both female and male geckos (Figure 2C–F).

The principal coordinates analysis (PCoA) plot, based on the Bray–Curtis distance matrix, showed that most samples were similar to each other without clear segregation into independent groups (Figure 4A).

### 3.3. Comparable Gut Bacterial Composition in Female and Male Geckos

The one-way PERMANOVA analysis revealed non-significant differences in bacterial composition between sexes (*p* = 0.51). Among the bacterial sequences analyzed, 481 ASVs were shared between both sexes, constituting 57.7% of the total ASVs identified. Additionally, 108 ASVs (13.0%) were exclusive to females, while 244 ASVs (29.3%) were unique to males (Figure 4B). Among these exclusive taxa, there were six bacterial phyla in the female group and nine bacterial phyla in the male group. The most representative phylum in both groups was Firmicutes, with Clostridia being the most abundant class. At the family level, Lachnospiraceae and Ruminococcaceae were most representative in both sexes. In addition, Bacteroidota, followed by Actinobacteria, were among the taxa that were unique to females. In males, on the other hand, we observed a higher representation of the phylum Tenericutes (class Mollicutes).

## 4. Discussion

In this study, we aimed to expand the knowledge of the gut microbiome of wild-caught South American leaf-toed geckos in a sex-dependent manner. Although some researchers have proposed that sex may influence the structural composition of the host’s gut microbiome by interacting with sex hormones and sex-specific immune responses [16,50,51], our results showed that there was no effect of sex on the diversity and composition of bacterial communities in the South American leaf-toed gecko. Similar to our finding, Kohl and collaborators (2017) did not detect an effect of sex on bacterial diversity or the relative abundance of any taxa between omnivorous and herbivorous lizards [11]. We only recorded differences between females and males in the abundance of the phylum Bacteroidota, where females presented a higher abundance of this group. Studies have shown that the gut microbiome composition in female lizards, including species like *Sceloporus virgatus*, *Rhinella marina*, and *Calotes versicolor*, is influenced by sex, captivity, and diet [17,52,53]. In this context, the higher abundance of the phylum Bacteroidota in female lizards is indeed related to their diet [52]. The phylum Bacteroidota are major members of the animal gut microbiome [54], playing functional roles in degrading high-molecular-weight organic matter (i.e., proteins and carbohydrates), activating T-cell-mediated responses, and producing butyrate to maintain gut health [55,56]. Additionally, members of Bacteroidota display intricate mechanisms for acquiring and breaking down dietary polysaccharides that are otherwise difficult to digest, such as chitin [56]. Chitin, one of the most abundant biopolymers in nature, is present in the diets of many vertebrates [57], including *P. gerropygus*. Prey items typically consumed by the South American leaf-toed gecko in the Atacama Desert primarily consist of arthropods, particularly Coleoptera and insect larvae [39], which are rich sources of chitin. Chitin can potentially restore the microbial community’s compositional balance, showing promising anti-viral, anti-tumor, antifungal, and antimicrobial effects [57,58]. Thus, the enzymatic and regulatory activities of Bacteroidota may contribute to reptile adaptation to digesting more chitin-rich prey items [58,59] in a sex-dependent way.

On the other hand, the ratio of Firmicutes to Bacteroidota (F/B) was significantly higher in males than in females. Some authors have proposed that an increased Firmicutes to Bacteroidetes (F/B) ratio in the gut microbiome indicates a greater energy harvesting capacity for hosts [60,61]. For example, Tang et al. (2022) found that the F/B ratio in the gut of wild Tokay geckos (*Gekko gecko*) was higher than that of captive animals, suggesting that higher F/B ratios in wild animals help digest and absorb food nutrients more efficiently, allowing the host to obtain energy in wild populations [62]. Firmicutes also contribute to fiber and cellulose degradation by breaking down cellulose into volatile fatty acids, which can be used by the host for the degradation of fiber and cellulose [52]. Moreover, the family Lachnospiraceae within Firmicutes is recognized for its involvement in chitin hydrolysis [63]. Likewise, Ruminococcaceae bacteria have been implicated in chitin digestion within the gut microbiome of insectivorous mammals [64,65]. The variations in abundance levels observed in specific groups between female and male geckos suggest a potential adaptive strategy for wild geckos surviving in harsh natural environments. Further investigation into these differences may elucidate how each sex adapts to distinct environmental and physiological challenges.

Our study showed that the phyla Bacteroidota, Firmicutes, and Proteobacteria accounted for more than 97% of the gut microbiome, indicating that these bacterial phyla were the predominant ones in *P. gerrhopygus*. In contrast to our results, the top four dominant bacterial phyla in the Japanese gecko (*Gekko japonicus*) and leopard gecko (*Eublepharis macularius*) were Verrucomicrobiota, Bacteroidota, Firmicutes, and Proteobacteria [11,57]. It was recently reported that Proteobacteria, Firmicutes, Bacteroidota, and Actinobacteria were the dominant phyla of the gut microbiome of the Tokay gecko (*Gekko gecko*) [62]. However, our results are consistent with observations in other reptilian taxa, where the dominant gut microbial phyla were Bacteroidota, Firmicutes, and Proteobacteria [11,52,66,67,68].

The phylum Proteobacteria, which was the third most abundant in our samples, is often associated with dysbiosis, a sign of gastrointestinal tract disease in animals, including humans [69]. However, Proteobacteria also play a crucial role in maintaining gut pH, degrading and fermenting complex sugars, and producing vitamins for their hosts [66,70]. Notably, members of the family Desulfovibrionaceae are known for their ability to reduce sulfate, producing hydrogen sulfide (H_2_S) as a byproduct. This process is vital for maintaining a balanced gut environment and influencing several physiological processes, including modulating inflammation and maintaining epithelial integrity, as well as influencing host immune responses [50,71].

At the family level, our samples were predominantly characterized by Bacteroidaceae, Lachnospiraceae, and Porphyromonadaceae, with lesser contributions from Ruminococcaceae and Helicobacteraceae. This finding is consistent with previous findings on reptile gut microbiomes. For instance, Arizza et al. (2019) found that the dominant bacterial families in the gut microbiome of sea turtles (*Caretta caretta*) included Ruminococcaceae, Rikenellaceae, Lachnospiraceae, and Clostridiales [66]. However, our results differed from those reported for other geckos, such as the wild-caught Japanese gecko (*G. japonicas*), where the dominant bacterial families were Akkermansiaceae, Bacteroidaceae, Tannerellaceae, Enterobacteriaceae, Lachnospiraceae, and Clostridiaceae [57]. These discrepancies are likely due to differences in the species, the environmental conditions, and the sampling methods.

At the level of composition of unique bacterial taxa for female and male geckos, we observed a high representativity of Tenericutes (class Mollicutes) in males. The Tenericutes members have been identified as important members of gut communities in fish, amphibians, reptiles, and mammals, which may exert specific roles in nutrient processing [70]. For example, in the study of Zhou et al. (2020), the northern grass lizard (*Takydromus septentrionalis*) tends to exhibit a higher proportion of Tenericutes compared to their wild counterparts, which might be related to the limited activity space and food sources of lizards [72]. Although our study was conducted on geckos in their natural environment, some studies have reported that male geckos (*Hemidactylus frenatus*) displayed higher levels of aggression and territorial behavior than females, which led to reduced mobility. Additionally, male geckos may need to stop and fight more often than females due to their territorial nature and aggressive tendencies [73]. Related to this observation, the structure of gut microbial communities in vertebrates may influence behavioral aspects such as aggressiveness [74]. Further research into the behavior of this species would help better understand the functional contributions of Tenericutes in the gut of male geckos.

Finally, our data indicate that 9% of the total bacterial families still need to be identified, revealing that many classes and their metabolic capabilities are still to be unveiled.

## 5. Conclusions

The present study provides valuable insights into the gut microbiome composition and abundance in wild-caught South American leaf-toed geckos (*Phyllodactylus gerropygus*). Our analysis revealed that the bacterial phyla Bacteroidota, Firmicutes, and Proteobacteria constitute the predominant core microbiome of *P. gerropygus*. Notably, significant sex differences were observed in the abundance of the Bacteroidota group at various taxonomic levels in females, while males exhibited variations in the Firmicutes/Bacteroidota ratio. Although certain taxa displayed higher abundance in specific sexes, these differences did not reach statistical significance. These findings suggest that females and males exhibit variations in the types of food items consumed in their diet. Importantly, our study sets the stage for future investigations aiming to elucidate the impact of gut microbial communities on the ecological dynamics of species inhabiting highly unpredictable environments, leading us to explore more complex ecological niches, such as those in the hyperarid Atacama Desert.

## Figures and Tables

**Figure 1 microorganisms-12-01194-f001:**
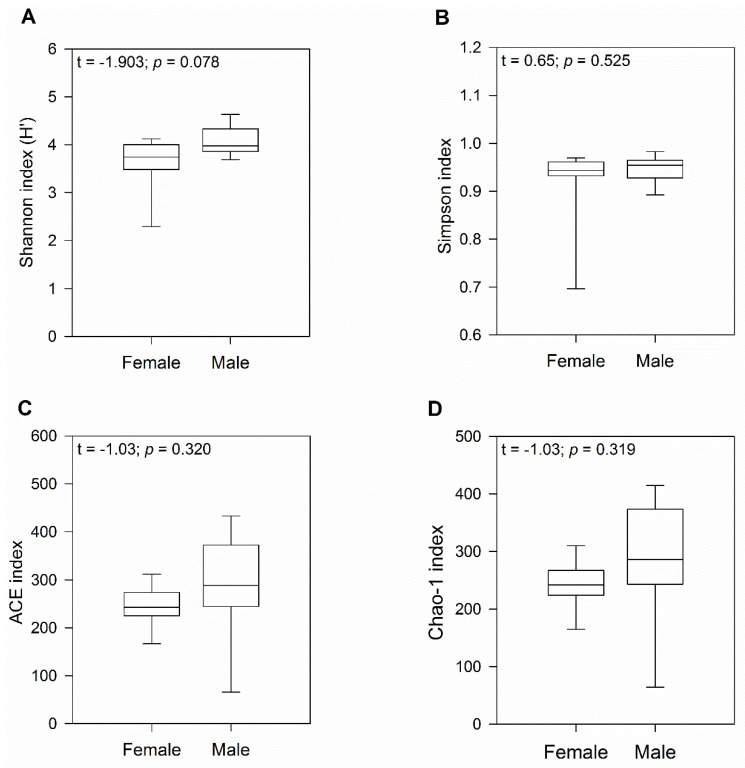
Diversity and gut microbiome richness in wild-caught South American leaf-toed geckos (*Phyllodactylus gerropygus*). Boxplot of the (**A**) Shannon (H′), (**B**) Simpson diversity, (**C**) ACE, and (**D**) Chao-1 richness indices. The t value corresponds to the Student’s *t*-test.

**Figure 2 microorganisms-12-01194-f002:**
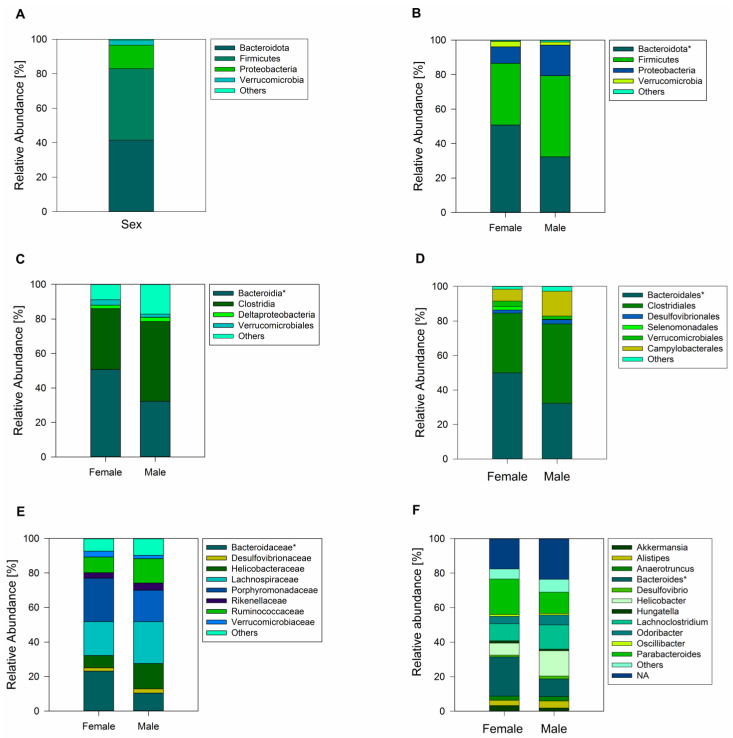
Analysis of the gut microbiome of the South American leaf-toed gecko. (**A**) Total relative abundance (%) of fecal bacterial communities at the phyla level across sexes. Relative abundance of the gut microbiome for female and male geckos at the phyla level (**B**), class (**C**), order (**D**), family (**E**), and (**F**) family. Only phyla, class, order, family, and gender with relative abundance greater than 1% are shown in the histogram, and the other taxa are combined (others). The asterisk next to the respective name indicates statistically significant differences between females and males.

**Figure 3 microorganisms-12-01194-f003:**
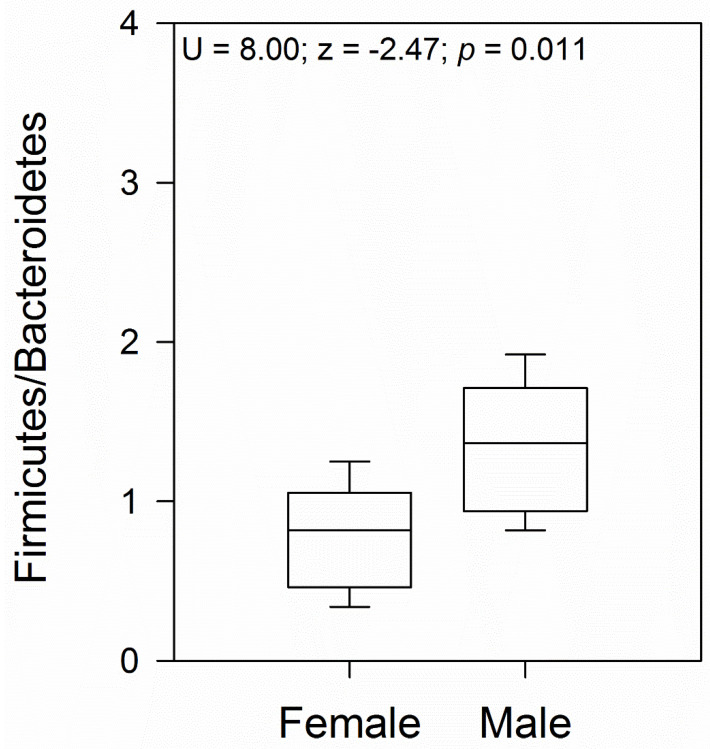
The ratio of Firmicutes to Bacteroidetes (F/B) boxplot for female and male geckos. The U and z values correspond to the Mann–Whitney test.

**Figure 4 microorganisms-12-01194-f004:**
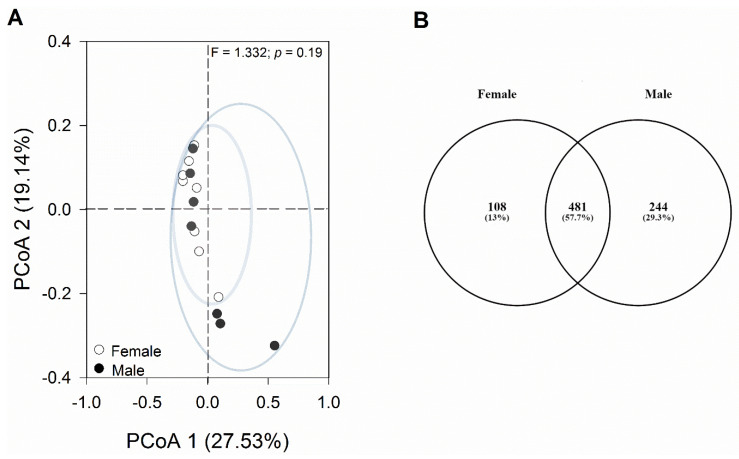
Analyses of changes in bacterial composition. (**A**) Principal coordinates analysis (PCoA) was conducted on the Bray–Curtis distance matrix to assess the diversity of bacterial communities in female and male geckos. Each data point on the graph corresponds to one sample, with different colors denoting each sex. (**B**) Venn diagram of core and specific ASVs of female and male gecko fecal samples. Each ellipse represents a group. The number of ASVs shared among all the groups is shown in the center, and the number of specific ASVs is shown in the non-overlapping proportions of each ellipse.

## Data Availability

The raw data supporting the conclusions of this article will be made available by the authors on request.

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
