# Peer review of "Insights into the Gut Microbiome of the South American Leaf-Toed Gecko (Phylodactylus gerropygus) Inhabiting the Core of the Atacama Desert"

_microorganisms, 2024, doi:10.3390/microorganisms12061194_

Round 1
Reviewer 1 Report
Comments and Suggestions for Authors
The manuscript discussed gut microbiome of the South American Leaf Toed Gecko (Phylodactylus gerropygus) inhabiting the core of the Atacama Desert.
There are a few points that can improve the presentation.
Abstract: Needs to be revised and organized. Add P value. Have a solid conclusion.
L110: 3 ml/year? is this correct?
L115: was age taken into consideration? were the samples pooled?
Fig 2. Use bigger font for the key.
Discussion: discuss how diet is different between sexes and how does this impact microbiota.
Conclusion: revise and have a solid conclusion based on your results, don't use references.
Author Response
Abstract: Needs to be revised and organized. Add P value. Have a solid conclusion.
Response: We thank the reviewer for his comments and suggestions. In this new version, we have rewritten the Abstract to be more consistent with the results reported in the manuscript. All changes have been highlighted in yellow for easy identification.
L110: 3 ml/year? is this correct?
Response: We correct this information, see Lines 137-138.
L115: was age taken into consideration? were the samples pooled?
Response: We only considered sampling adult individuals, and to determine whether they were juveniles or adults we used the snout-vent length = 49 ± 7 mm. We make this clarification in Lines 143-147.
Fig 2. Use bigger font for the key.
Response: We redrew Figure 2 and increased the font size in the captions.
Discussion: discuss how diet is different between sexes and how does this impact microbiota.
Response: We thank the reviewer for this suggestion. In the new version of the manuscript we further develop the discussion of how diet differs between sexes. Lines 277-314
Conclusion: revise and have a solid conclusion based on your results, don't use references.
Response: We thank the reviewer for this observation. In the new version of the manuscript, we have revised the conclusions and rewritten them to be more consistent with the results reported in the manuscript. All changes have been highlighted in yellow for easy identification.

Reviewer 2 Report
Comments and Suggestions for Authors
The gut microbiome plays a vital role in vertebrate species' health, nutrition, and adaptation, especially those living in extreme environments. This study aimed to characterize the composition and abundance of the gut microbiome in wild-caught South American leaf-toed geckos inhabiting the hyperarid Atacama Desert in northern Chile. It is worthy to publish this contribution. However, before it is formally accepted, there are some issues needed authors to make it clear.
1. 25-26 lines. What is the relationship between diet and chitin? Is the diet different for females and males? What are the dietary differences please elaborate.
2. 26-27 lines. Tenericutes, class Mollicutes are in high abundance, what are the functions of these bacteria and what role can they play in the gut.
3. 26-27 lines. Male phyla Tenericutes, class Mollicutes are high in abundance and are not represented in Figure 2.
4. The link between gender and gut microbes is absent in background of the introduction. Why does the significance of the research in this paper fall on the effect of gender on gut microbes? Suggest additional clarification
5. 121 lines, why record the weight of geckos? It is not used in the analysis. And there is only a left half bracket in the sentence, without the right side.
6. 133 lines, Unit μl is changed to μL.
7. Does need italics for the later of RNAlater in line 134? Some other literature does not have italics.
8. 180-181 lines. “unchanged”? Maybe not significantly different.
9. In results, Why is there no rarefaction curve and is the sequencing depth sufficient?
10. Lines 182-183, Bacterial α- Diversity only use the Shannon index for diversity is a bit simplistic, α- Diversity includes species diversity and species richness, which can be comprehensively analyzed using multiple indicators, such as ACE and Chao1, etc.
11. Line 193, Bacteroidota should not start a new paragraph.
12. There is an inconsistency in the description of the results in lines 26-27 and 194-197.
13. Lines 236-239, bacterial genus names need to be italicized.
14. The description of this sentence in lines 231-232 does not seem to match (D) in Figure 2. At the family level, Lachnospiraceae and Ruminococcaceae were most abundant in female geckos, while a reverse pattern was observed in males.
15. 253-236 lines. The description is inaccurate and the microbial classification of male and female geckos at the genus level is cross-cutting.
16. In Figure 2. Few of the microbial descriptions show the "phylum" level of classification and why there are no descriptions at the genus level of classification. The microbial diversity covered in the article should be analysed and discussed at the genus level, which is more meaningful.
Author Response
Below you will find the answers to each of the questions and suggestions that the reviewer made.
Comments and Suggestions for Authors
The gut microbiome plays a vital role in vertebrate species' health, nutrition, and adaptation, especially those living in extreme environments. This study aimed to characterize the composition and abundance of the gut microbiome in wild-caught South American leaf-toed geckos inhabiting the hyperarid Atacama Desert in northern Chile. It is worthy to publish this contribution. However, before it is formally accepted, there are some issues needed authors to make it clear.
We would like to thank the reviewer for his comments and suggestions. We believe that the feedback helped us to improve our manuscript. We have responded to each of your questions and all changes in the text are underlined in yellow
- 25-26 lines. What is the relationship between diet and chitin? Is the diet different for females and males? What are the dietary differences please elaborate.
Response: We thank the reviewer for this comment. We have rewritten the Abstract section. However, in this new version of the manuscript, we have added in the Discussion section the relationship between diet and chitin and how it can be influenced by sex. Lines 281-312.
- 26-27 lines. Tenericutes, class Mollicutes are in high abundance, what are the functions of these bacteria and what role can they play in the gut.
Response: We thank the reviewer for this comment. We have rewritten the Abstract section. The Tenericutes members have been identified as important members of gut communities in fish, amphibians, reptiles, and mammals, which may exert specific roles in nutrient processing (Colston & Jackson, 2016). However, the exact functional roles and ecological significance of the diverse, uncultured Tenericutes lineages in the gut microbiome are still largely unknown and require further investigation.
- 26-27 lines. Male phyla Tenericutes, class Mollicutes are high in abundance and are not represented in Figure 2.
Response: We thank the reviewer for this comment. We have rewritten the Abstract section.
- The link between gender and gut microbes is absent in background of the introduction. Why does the significance of the research in this paper fall on the effect of gender on gut microbes? Suggest additional clarification
Response; In this revised version, we provide clarification regarding our analysis of species composition using Venny analysis (Figure 4B). We noted that within the unique bacterial groups identified in females and males, the Firmicutes phylum was predominant. Notably, in males, the phylum Tenericutes, specifically the class Mollicutes, was the second prominent group.
- 121 lines, why record the weight of geckos? It is not used in the analysis. And there is only a left half bracket in the sentence, without the right side.
Response: We thank the reviewer for noting that. Effectively the weight of geckos was not used in this study. We eliminate this observation across the methodology
- 133 lines, Unit μl is changed to μL.
Response: It has been corrected.
- Does need italics for the later of RNAlater in line 134? Some other literature does not have italics.
Response: We agree and have corrected it
- 180-181 lines. “unchanged”? Maybe not significantly different.
Response: We thank the reviewer for observing that. In this new version, we corrected it
- In results, Why is there no rarefaction curve and is the sequencing depth sufficient?
Response: We thank the reviewer for this observation. In this revised version, we provide the rarefaction curve. It is included in the supplementary material (Fig. S1).
- Lines 182-183, Bacterial α- Diversity only use the Shannon index for diversity is a bit simplistic, α- Diversity includes species diversity and species richness, which can be comprehensively analyzed using multiple indicators, such as ACE and Chao1, etc.
Response: We thank the reviewer for this observation. We agree with it. In this new version, two richness indices (ACE and Chao1) and two diversity indices (Shannon and Simpson) have been included.
- Line 193, Bacteroidota should not start a new paragraph.
Response: We agree and have corrected it.
- There is an inconsistency in the description of the results in lines 26-27 and 194-197.
Response: We appreciate the reviewer's feedback and have thoroughly revised both the Results and Discussion sections of the manuscript.
- Lines 236-239, bacterial genus names need to be italicized.
Response: We agree and have corrected it.
- The description of this sentence in lines 231-232 does not seem to match (D) in Figure 2. At the family level, Lachnospiraceae and Ruminococcaceae were most abundant in female geckos, while a reverse pattern was observed in males.
Response: This section encompasses ASVs exclusive to either females or males following analysis with Venny. Here, we specifically highlight phyla, classes, and families, that exhibit greater representativeness in females and males. We acknowledge our error in referring to "abundance" and have corrected it to "representativeness." In fact, the observation that the phylum Tenericutes (class Mollicutes) is present in males is based on this analysis. In this new version of the manuscript, the way the results are presented has been rephrased and modified.
- 253-236 lines. The description is inaccurate and the microbial classification of male and female geckos at the genus level is cross-cutting.
Response: We agree and have corrected it.
- In Figure 2. Few of the microbial descriptions show the "phylum" level of classification and why there are no descriptions at the genus level of classification. The microbial diversity covered in the article should be analysed and discussed at the genus level, which is more meaningful.
Response: We thank the reviewer for this observation. We agree with it. In this new version of the manuscript, Figure 2 has been redesigned to include abundances by genera whose abundance was greater than 1%.

Reviewer 3 Report
Comments and Suggestions for Authors
Author Response
We would like to thank the reviewer for his valuable comments and appreciation. In response to the reviewer's feedback, we have revised the conclusions of the manuscript to be more consistent with the results reported in the manuscript. All changes have been highlighted in yellow for easy identification. Please see the attachment.
